# Validation of a finite-element model of a 5 m three-row roller wind turbine blade bearing

Matthis Graßmann[1], Martin Geibel[1], Florian Schleich[1]

[1] Fraunhofer Institute for Wind Energy Systems IWES, Large Bearing Laboratory, Am Schleusengraben 22, 21029 Hamburg, Germany

*Correspondence to*: Matthis Graßmann (matthis.grassmann@iwes.fraunhofer.de)

**Abstract.** Large rolling bearings with complex interfaces need reliable finite-element models to determine the load distribution and deformation behavior. To ensure the accuracy of the results, it is important to validate the models against experimental data. Several works on models with different approaches are published but rarely is this validated. The present work now firstly validates a finite-element model of an original size three-row roller wind turbine blade bearing. For the validation, strain gauges and laser sensors are used to compare the deformation behavior and radial displacements of the bearing rings against experimental results. A characteristic of three-row roller bearings is the segmentation of one of the rings for manufacturing purposes. In this work, the authors investigate the influence of different coefficients of friction between the segmented outer ring and different bolt preloads on the occurring strain on the bearing rings. Two different sets of bolt preloads were considered: One to represent operational behavior with no relative movement between the segments of the split ring and one with gap opening and sliding to investigate nonlinear behavior of the bearing. The result of this work is a validated finite-element bearing and test rig model for different parameter sets and loads.

## 1 Introduction

Blade bearings connect the rotor blades of a wind turbine to the rotor hub and enable the rotation (pitching) of the blades along their longitudinal axis. Pitching controls the wind turbine's power output and acting loads on the blades and blade bearings by adjusting the position of the blades in the wind. Blade bearings typically are four-point contact ball bearings or roller bearings with several meters in diameter and many rolling bodies (Hau 2016; Stammler et al. 2024). Due to inhomogeneous stiffnesses of the blade flange and hub and high alternating loads, the internal load distribution of a blade bearing is important for its calculation and design process. Finite-element (FE) simulations are the only means to consider these aspects sufficiently. The validation of FE models against experimental data is most important to ensure the reliability of the simulation results (ASME 2019).

Different approaches for modelling roller bearings with various dimensions have been published. DEMIRHAN et al. (2008) simulated two-dimensional FE bearing models with outer diameters between 85 mm and 250 mm with solid modelled rollers. In the FE, they considered rigid bearing rings to compare the load-deformation behavior with analytical models and elastic

rings to compare the bearing stiffness against experimental results. HAO et al. (2019) also simulated a two-dimensional cylindrical roller bearing model with an outer diameter of 100 mm. They compared the resulting radial deformation of the bearing model against experimental data and analytic results. They obtained maximum deviations between FE model and experiment of 7.5% and 42.5% between analytical calculation and experiment. WANG et al. (2017) and HE et al. (2018) simulated a three-row roller bearing model with a pitch diameter of nearly 1 m. They ran an experiment where they compressed

a single roller between two plates to verify their simplified FE model of a solid roller by comparing the deformation. Then they compared the results of their bearing model with different numbers of springs for each roller against their solid roller model. They showed significant differences in the load distribution between the spring and the solid models when they use less than three springs and achieved less deviation the more springs they use. However, they do not considered any surrounding structures of the bearings for their simulation and they did not validate their entire bearing model. BECKER et al. (2014)

simulated a tapered roller bearing as the main bearing of a 6 MW wind turbine considering the housing of the gearbox and generator and hub. They divided the rollers into segments along their length and modelled each segment using a combination of a nonlinear spring and gap elements. They showed the influence of the bolt preloads and the operating loads on the roller loads in the bearing. However, they do not validated their FE bearing model against experimental data. STAMMLER et al. (2018) simulated a 5 m three-row roller bearing as blade bearing with focus on the resulting load distribution. They also modelled the

rollers with nonlinear spring elements. In their simulations they considered the surrounding structures hub and blade and highlighted their influence on the bearing behavior compared to stiff interfaces. Furthermore, they investigated the influence of different pitch and load angles on the load distribution. However, they do not validated their FE bearing model against experimental data.

For ball bearings different modeling approaches with focus on obtaining the load distribution of the bearings have been

published. They all use nonlinear springs to represent the ball-raceway interaction and form a basis for modelling ball bearings for the following publications. However, none of their models are validated (Daidié et al. 2008; Gao et al. 2011; Smolnicki and Rusiński 2007). CHEN et al. investigated the influence of different configurations of generic supporting structures and a realistic hub and blade root of a 1.5 MW wind turbine as well as modelled bolts with various preloads on the load distribution of a bearing. They highlighted significant differences between rigid bearing rings and deformable supporting structures.

However, they do not compared their results against any experimental data (Chen et al. 2017; Chen and Wen 2012). SCHWACK et al. (2016) simulated a double-row four-point contact ball bearing of the IWT7.5-164 reference wind turbine including its hub and a simplified blade flange. They focused on the load distribution and contact angle deviation as well as the resulting stress distribution in the blade bearing for different load cases. They do not validated their bearing model against experimental data. SCHLEICH et al. (2024) simulated the double-row four-point contact ball bearing of the IWT7.5-164 reference wind

turbine as well. They focused on reducing single components of the rotor by means of super-elements and investigated the differences between a full and a one-third rotor model. They concluded their work with the necessity of considering a full rotor model with three individual loaded blades to obtain realistic blade bearing loads for further calculations. They do not validated their bearing model. LIU et al. (2018) simulated a single-row four-point contact ball bearing with a pitch diameter of 1 m. They

used a test rig with simple structural steel components with homogenously distributed high stiffness to apply a bending moment to the bearing using two hydraulic cylinders. They compared the deformation of the bearing rings from the FE model with the experimental results using strain gauges. For the maximum strain, they obtained deviations from 10.7% up to 26.9% depending on the load level. In previous publications, the authors of the present work have successfully validated the FE model of a double-row four-point contact ball bearing with an outer diameter of 750 mm against experimental data using a blade bearing test rig. They used strain gauges on the surface of the outer bearing ring to compare the deformation behavior of the bearing model and the experiment in tangential and axial direction. Their deviation for the maximum occurring strain was less than 10% (Graßmann et al. 2023; Graßmann et al. 2024).

The authors have defined two criteria for the validation of finite-element models of slewing bearings, namely:

- For the maximum strain, the FE results should deviate less than 10 % from the experimental data.
- The characteristic course (number and position of maxima and minima) of the FE results should match the experimental mean values.

There is no validation of large-scale roller bearings. This work will fill that gap. The methodology including the bearing and used test rig as well as their FE models are described in Section 2. In Section 3, the results of the tests and the validation of the FE bearing model for different sets of bolt preloads are discussed. Section 4 concludes the work.

## 2 Methodology

### 2.1 Bearing

The bearing in this paper is a grease-lubricated three-row roller bearing with an outer diameter of nearly five meters. The bearing rings are mounted to surrounding structures through bolts. In a wind turbine, the outer ring would be typically mounted to the hub and the inner ring to the blade. The bearing has two rows of axial rollers that carry the larger axial component and one row of radial rollers that carry the smaller radial component of external loads. Figure 1 shows a cross-sectional schematic view of the bearing.

For manufacturing and assembling purposes, the outer ring is segmented into a lower and upper part to place the rolling elements on the raceways. The outer ring bolt forces thus largely determine the preload of the axial rolling elements. Further, the outer ring split can potentially open or have relative movement between the two surfaces when the bolt forces are too small. Especially a gap opening can be critical for further operation, as lubricants can enter the gap, find its way to the bolts and there reduce friction in the threads, which in turn can cause bolt loosening and catastrophic failures.

Under operation, the bearing rings deform which increases the risk of high contact pressures on the edge of the rollers. To counteract edge loading, the rollers of large roller bearings typically are manufactured with a profile to ensure a more even pressure distribution along the rotational axis of the individual rollers. The rollers in the bearing of this work have a standard logarithmic profile according to ISO/TS 16281 (2008). Table 1 lists the main dimensions of the bearing.

 **Table 1: Main dimensions of the bearing**

| Property | Symbol | Value | Unit |
|---|---|---|---|
| Inner diameter | - | 4390 | mm |
| Outer diameter | - | 4972 | mm |
| Pitch diameter axial rows | $D_{pw,axial}$ | 4714 | mm |
| Pitch diameter radial row | $D_{pw,radial}$ | 4798 | mm |
| Inner bolt circle diameter | - | 4500 | mm |
| Outer bolt circle diameter | - | 4880 | mm |
| Inner ring bolts | - | 120 | - |
| Outer ring bolts | - | 120 | - |
| Total height | - | 298 | mm |
| Axial rollers per row | - | 124 | - |
| Diameter axial rollers | $D_{we,axial}$ | 50 | mm |
| Effective length axial rollers | $L_{we,axial}$ | 47.5 | mm |
| Radial rollers | - | 568 | - |
| Diameter radial rollers | $D_{we,radial}$ | 20 | mm |
| Effective length radial rollers | $L_{we,radial}$ | 18 | mm |
| Mass | $m$ | 8013 | kg |

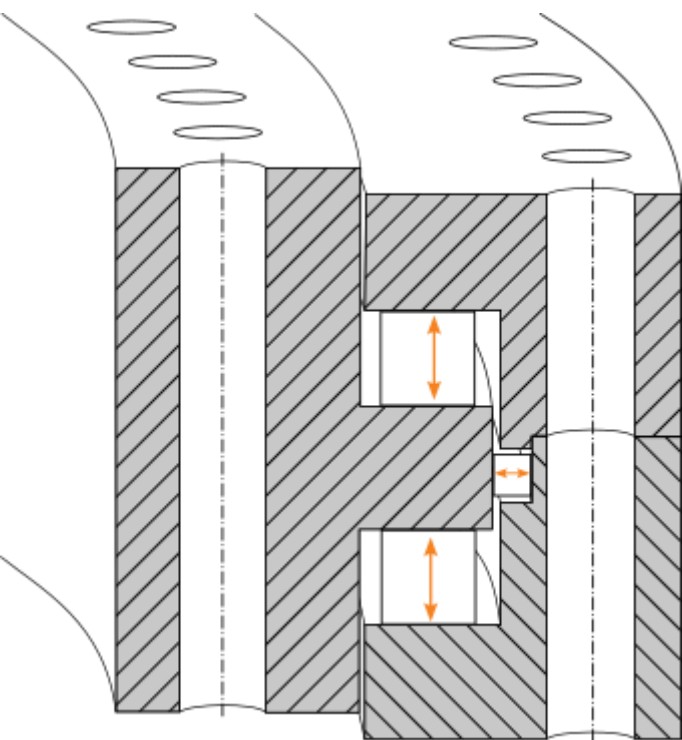

**Figure 1: Cross-sectional view of a three-row roller bearing with a segmented outer ring**

## 2.2 FE bearing model

The bearing in this work contains a total number of 816 rollers. To save computational time, the rollers are modelled with nonlinear spring elements. Five springs represent one axial roller, and three springs represent one radial roller. Force distributed constraints (FDCs) connect the spring elements to the raceways. Internal investigations have shown no significant differences regarding the roller forces and ring deformation when the rollers are modelled with five spring elements and 31 spring elements. Figure 2 shows the modelling of the rollers. The spring elements are displayed in orange, the connected parts of the

raceways in green and the FDCs are indicated with dashed blue lines. Each raceway is divided into segments along their circumference according to the number of the rollers. These segments are then further divided into segments in radial direction according to the number of spring elements that represent one roller. That allows for connecting each spring with sole their belonging part of the raceways.

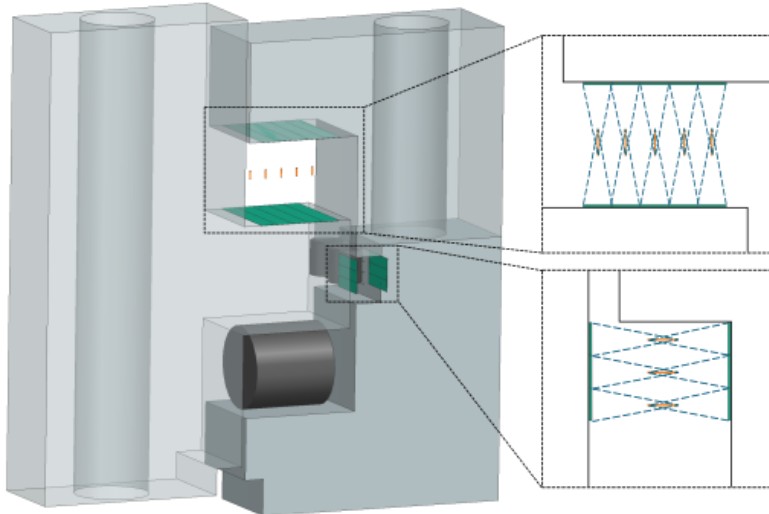

**Figure 2: Modelling the rollers of a three-row roller bearing with nonlinear springs**

The nonlinear behavior of the spring elements is controlled by a force-deformation curve that is based on analytical calculations of the stiffness and the number of springs per roller. The stiffness of the roller-raceway contact can be calculated depending on the effective roller length (Palmgren 1964; DIN 26281 2010).

Because of the rather complex geometry of the bearing rings, they are further segmented to control the FE mesh. Bonded contacts virtually glue these segments together. Figure 3 shows the segmentations of the bearing rings on the left and the resulting mesh on the right. The orange lines indicate segmentations that allow a controlled meshing of the rings. The blue lines highlight the contact between the two outer rings. In the FE model, this is modelled as a frictional contact with various coefficients of friction between 0.1 and 0.2 to investigate its influence on the deformation behavior of the bearing (cf. Section 3). All bonded contacts in this work use the MPC formulation while the frictional contacts use the Augmented Lagrange formulation. For all contacts, the default values for contact stiffness and penetration tolerance are used.

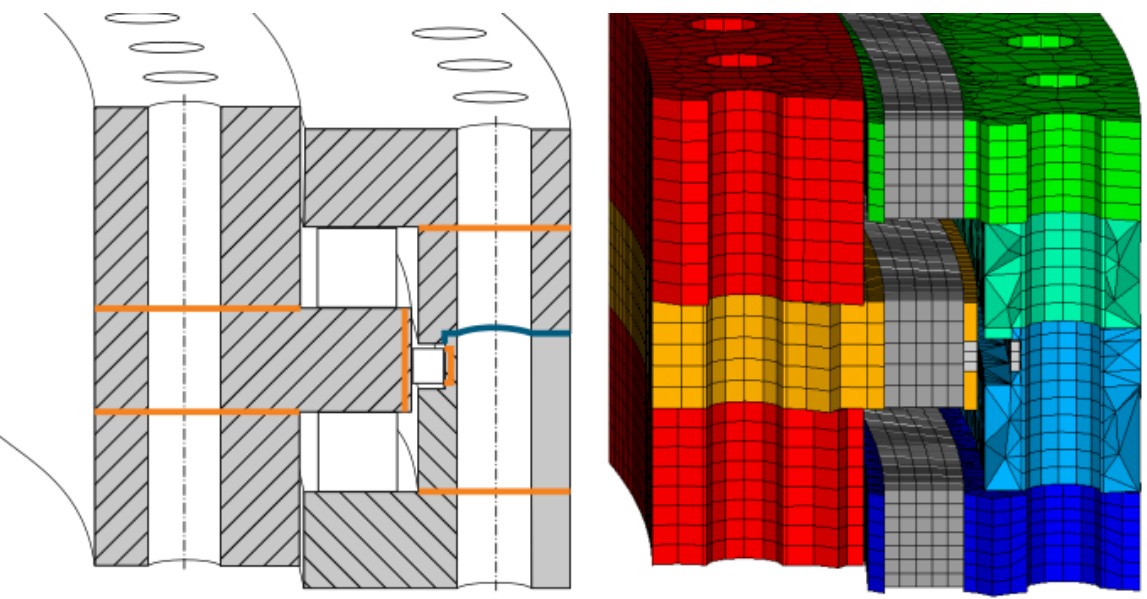

**Figure 3: Defined contacts between the bearing ring sections (left) and resulting mesh of the bearing rings (right)**

### 2.3 Test environment

The test rig used in this work is the Fraunhofer IWES BEAT6.1 (Bearing Endurance and Acceptance Test rig). The rig tests two bearings with diameters up to six meters simultaneously. Six hydraulic cylinders in a hexapod configuration can apply loads in any degree of freedom through a steel structure called load platform with a maximum static bending moment of 50 MNm. An additional hydraulic cylinder connects to the inner ring of the lower bearing to apply any pitch movement. Calibrated load cells at the cylinders measure the applied loads. The interface parts, called hub adapters (HA) and force transition element (FTE) are designed to represent the stiffness of hub and blade of the Fraunhofer IWES reference wind turbine IWT7.5-164 (Popko and Thomas 2018). In the coordinate system of the blade bearing (according to Germanischer Lloyd (DNV GL AS 2016)), the hub connects to the main shaft at 0° which results in a locally stiffener region for the outer ring of the bearing in that area. Hence, the hub adapter that is used in the test rig is also stiffer around 0°. Figure 4 shows the test rig on the left and the bottom view and the cross section of the lower hub adapter with the stiffer segment at 0° on the x-axis. The used FTE is a hybrid FTE that consists of two glass fiber-reinforced plastics (GFRP) rings and steel segments in between. The bearings are mounted to the surrounding structures with bolts. The tightening of the bolts is realized with hydraulic tensioners in multiple steps. More detailed information about the test rig can be found in (Stammler 2020).

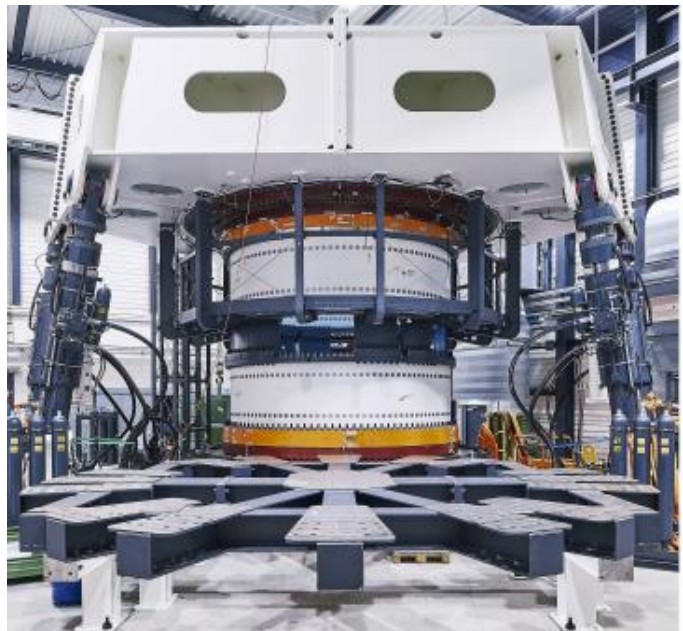 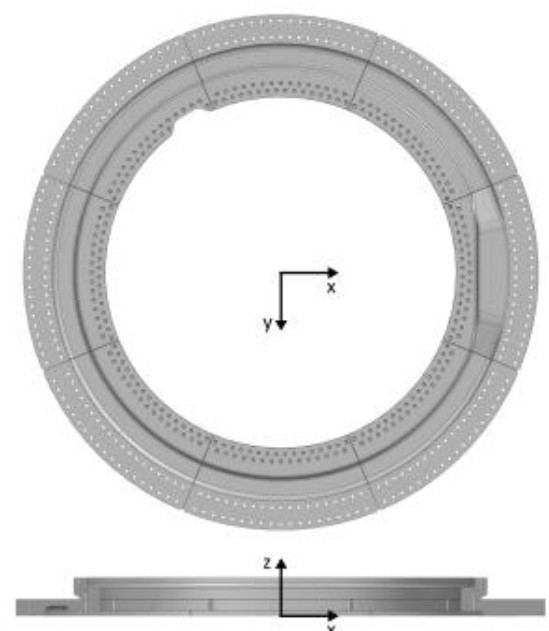

**Figure 4: Fraunhofer IWES blade bearing test rig BEAT6.1 (left) and bottom view and cross section of the lower hub adapter (right)**

The FE model of the test rig considers every component with frictional contacts between the flanges. The coefficient of friction is set to 0.2 for uncoated steel-to-steel contacts. The flanges of the bearings towards the surrounding structures are coated to increase the coefficient of friction to 0.67 for coated steel-to-steel contacts and 0.5 for coated steel-to-GFRP contacts. In total 1356 bolts are modelled as beam elements to mount all the components. To mount the bearings, nuts with frictional contacts between them and the bearing rings are modelled. The beam elements then are connected to the nuts or the threads of the FTE using FDC connections. The hydraulic cylinders are modelled as linear actuator elements which allow a similar load application as in the experiment. Figure 5 shows the FE test rig model as well as a more detailed view of the lower components and modelled bolts and nuts on the lower bearing. The reaction frame is the bottom white steel structure that connects the rig to the foundation. As boundary condition in the FE model, all degree of freedom of the bottom nodes of the reaction frame are locked. The top white steel structure is the load platform with attached hydraulic cylinders. The bearings are displayed in orange, the hub adapters in red. The FTE is shown in between the two bearings with the GFPR rings in white and steel segments in grey. The stiffener plates are mounted between the bearing inner rings and the FTE and displayed in yellow. The pitch support plate and adapters that connect the hydraulic pitch to the lower bearing are mounted to the bottom flange of the inner ring.

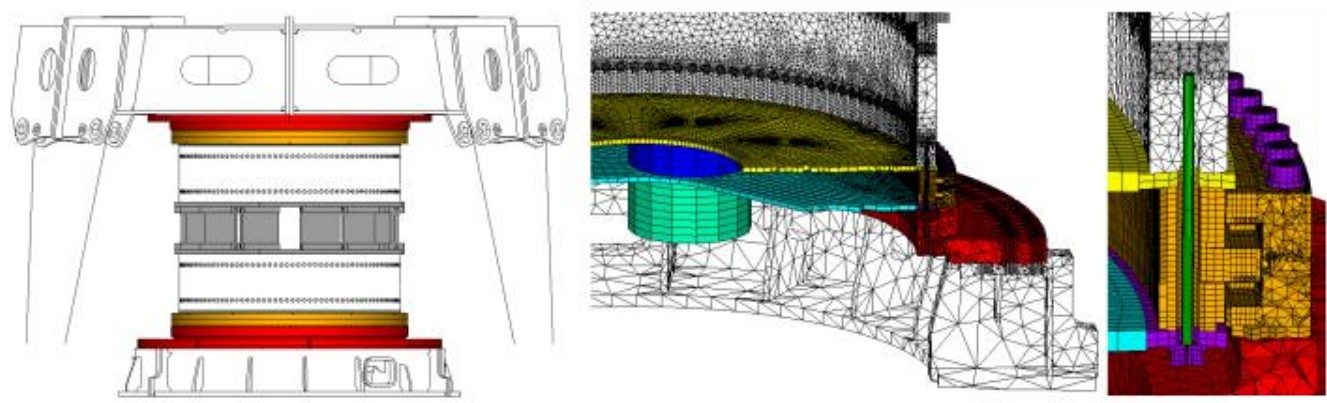

**Figure 5: FE model of the Fraunhofer IWES blade bearing test rig with detailed views on the lower components and modelled bolts**

In the simulation of the test rig, the first load step applies gravitational forces and bolts forces. The second load step applies load to the cylinders to compensate for the weight above the lower bearing. This is also done in the experiment and is used as starting point for every test as the lower bearing is unloaded. Following load steps then apply the load combinations that test the bearings.

## 2.4 Measurement

Strain gauges and laser sensors are used to validate the FE bearing model by comparing the simulation results against experimental data. Strain gauges are sensors that change their electrical resistance when they are expanded or compressed. Based on the change of electrical resistance, the change of the strain can be calculated. 24 strain gauges are glued to the lower bearing. 12 to the inner surface of the inner ring and 12 to the outer surface of the outer ring. On both rings, the strain gauges are placed each 30° to measure the axial and tangential strain at each position. All strain gauges are positioned near the free
flange of the rings and between the bore holes, because there the largest strain is expected. In the FE model, the strain gauges are modelled with shell elements that have the size of the foil that are glued to the bearing. The shell elements have no stiffness and are only used to evaluate occurring stress and strain and are placed at the same positions as in the experiment to allow a precise comparison. Each shell element is positioned on an underlaying element with exact the same size Figure 6 shows the positions of the strain gauges on the inner and outer ring of the lower bearing. It also shows one equipped strain gauge on the
outer ring with the bearing installed in the test rig.

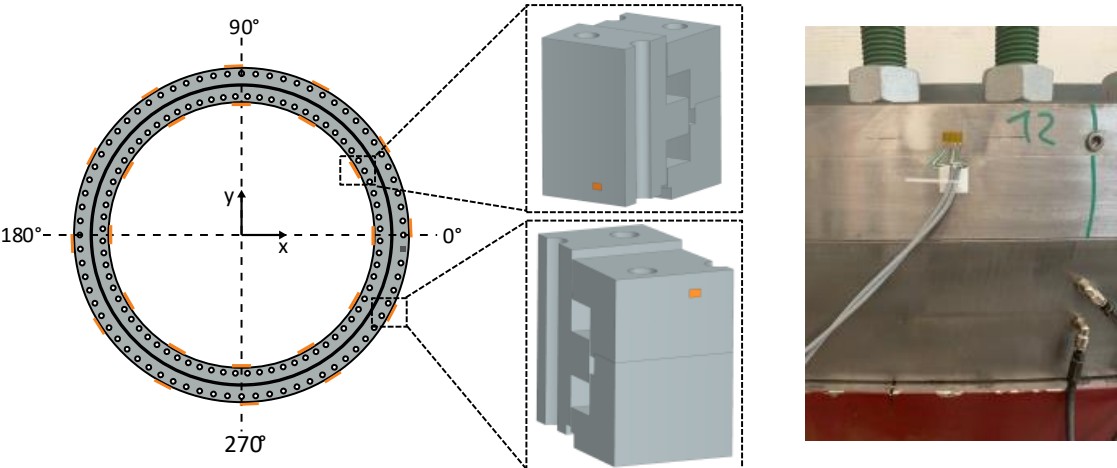

**Figure 6: Positions of the strain gauges on the inner and outer ring of the lower bearing**

Internal investigations with a validated FE model of a scaled blade bearing have shown significant influence on the occurring strain based on the alignment of the strain gauge. A translational misalignment of 3 mm and an angular misalignment of 2° have led to deviations of 15% of the maximum strain. Larger misalignments, especially of the angular position, have led to larger deviations. Besides the misalignment, the uncertainty of the strain gauges and the full measurement chain based on the Guide to the Expression of Uncertainty in Measurement (JCGM 2008) is less than 2%.

Laser sensors measure the distance between an object and the sensor. Two laser sensors at 0° and two sensors at 180° measure the radial displacement of the outer ring of the lower bearing in the highest loaded areas for bending moments around the y-axis. The sensors are installed at a lever arm that is fixed to the ground which allows for an absolute measurement of the ring displacement. At each position, one sensor aims at the center of the upper part of the split ring and one at the center of the lower part. As the laser sensors only measure a very tiny spot on the rings and the mesh of the bearing is quite course compared to the size of the laser point, multiple nodes that are located near the measured position are used to evaluate a mean displacement in the region of the laser measurement when the FE results are obtained. Figure 7 exemplary shows the laser sensors with highlighted measurement position on the outer ring of the lower bearing at 180°.

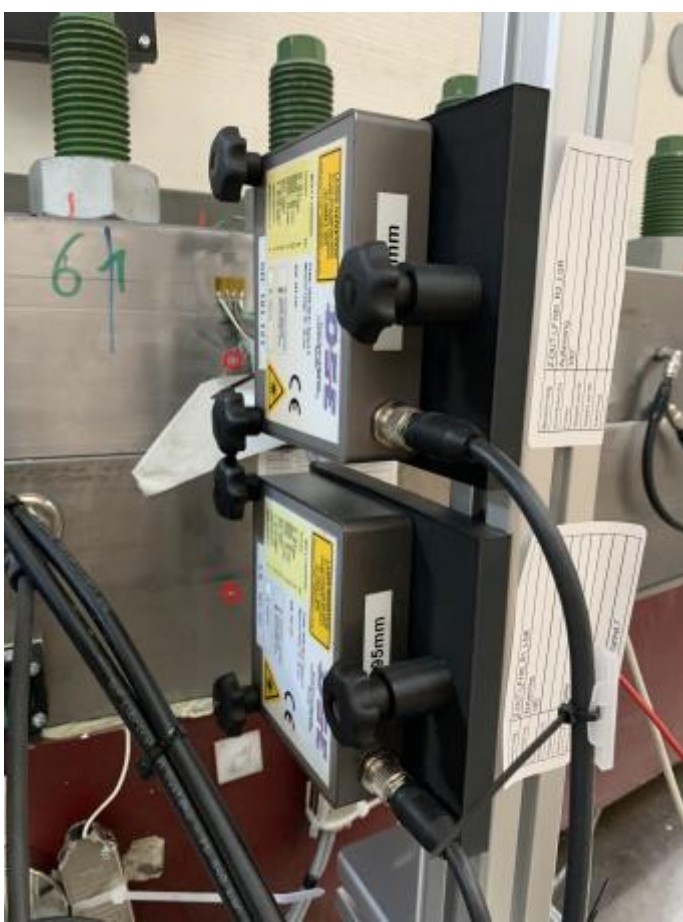

**Figure 7: Laser sensors to measure the radial displacement of the outer ring of the lower bearing at 180°**

## 2.5 Tests

Blade bearings are mounted to the surrounding structures using bolted connections. For three-row roller bearings, the bolt
preload at the segmented ring needs to be high enough to prevent a gap opening between the rings to ensure its reliability when
it is operating in the field. However, to investigate the behavior of the bearing with gap opening, two different sets of bolt
preloads for the lower bearing are tested. Calibrated measurement bolts that are equipped with strain gauges are used to
determine the actual bolt forces. At each bolt circle, 12 measurement bolts are distributed along the circumference. After the
bolt tightening process, the bolt forces are measured while the bearings are unloaded. Then the mean value and standard
deviation for each bolt circle is calculated. Table 2 lists the bolt preload sets that are used to mount the bearings to the test rig.
For both sets, the preload of the inner bolts that mount the inner rings of the bearings to the FTE remain constant. The bolt
preloads for the outer ring of the lower bearing are set to 330 kN for set 1 and 866 kN for set 2. Set 1 aims at forcing the gap
between the outer ring of the lower bearing to open to investigate how the bearing deformation changes due to nonlinear

behavior caused by sliding contacts. In contrast, set 2 aims at emulating realistic operating conditions without any sliding movements of the bearing rings.

As listed in Table 2, the standard deviation of the bolt preload differs between the bolt circles. In the test rig, the bolt preloads vary because of small inaccuracies of the tightening tools and the person who operates those tools. However, in the FE simulation all bolts are tightened with the same bolt preload. To analyze the influence of different bolt preloads on the behavior of the bearing, the standard deviation is added and subtracted to the mean bolt preload in the simulations. This is done for the lower bearing only, as it is of main interest.

**Table 2: Bolt preload sets to mount the lower bearing to the test rig including the standard deviation based on the bolt forces of the measurement bolts**

|  | Bolt forces inner ring in kN | Std bolt forces inner ring | Bolt forces outer ring in kN | Std bolt forces outer ring |
|---|---|---|---|---|
| 1 | 570 | 53 | 330 | 25 |
| 2 | 570 | 53 | 866 | 115 |

For the validation of the FE model, the simulated deformation of the bearing rings at the virtual strain gauges are compared against the experimental strain gauge data. For that, the test rig applies different load levels of static bending moments to the bearing, and the resulting strain is recorded. The test for bolt preload set 1 starts at $M_y = -30$ MNm and ramps up to $M_y = +30$ MNm in 1 MNm steps and holds every load level for 10 seconds without any pitch movements. For bolt preload set 2, the ramp starts at $M_y = -35$ MNm and goes up to $M_y = +35$MNm. For the comparison, the mean strain gauge value of the 10 seconds of each load step is used. To counteract differences in initial strain gauge values caused by the gluing process, the strain that occurs for an unloaded bearing is subtracted from the strain under load. For comparable reasons, this is also done for the simulation results.

## 3. Results and discussion

The following results compare the strain of the bearing rings at the strain gauge positions between the simulation and the experiment with the aim to validate the FE model. As the conditions like bolt preload (cf. Section 2.5) and frictional coefficient between the segmented bearing outer ring might vary in the experiment, different parameter sets are considered in the simulation. As described in Section 2.1, when gap opening of the outer ring split occurs, lubricants can enter the gap. That would entail different frictional conditions along the flange surfaces. Hence, coefficients of friction and bolt forces might be partially unknown. Therefore, both parameters vary in the simulations to find the best match to the measurements. The coefficient of friction for steel-to-steel contact is set to 0.2. Possible grease contamination in the contact area would reduce the friction. Hence, coefficients of friction of 0.15 and 0.1 are simulated. The coefficient of friction is kept constant for the entire

flange surface. In addition, the radial displacement of the outer ring of the lower bearing at 0° and 180° examined from the FE-model is compared with experimental results for bolt preload set 2.

## 3.1 Bolt preload set 1

Bolt preload set 1 investigates the behavior of the bearing when the preload of the bolted connections is too low and the gap between the segmented outer ring opens. Figure 8 shows the deformation of the bearing rings on the traction side for a negative bending moment of $My = -30$ MNm. According to the coordinate system in Figure 6, the traction side for a negative bending moment is at 0. It clearly shows a gap opening between the outer rings and small sliding of the upper outer ring.

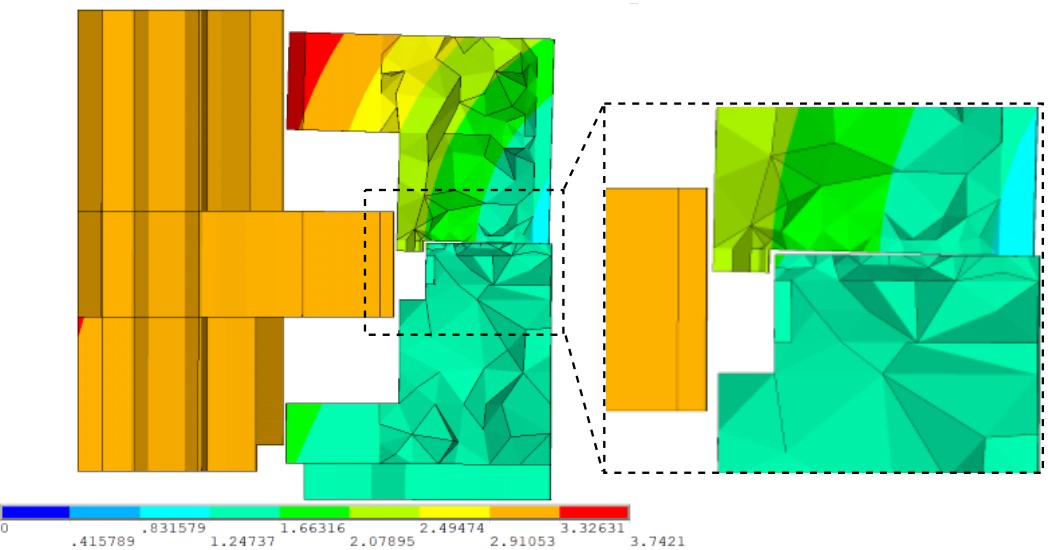

**Figure 8: Gap opening of the outer ring split of the lower bearing**

Figure 9 shows the axial and tangential strain of the lower bearing rings of a positive and negative bending moment of $My = \pm 30$ MNm for constant bolt preloads and different coefficients of friction between the outer ring segments. Each diagram displays the strain in µm/m on the y-axis over the circumferential position in degrees on the bearing rings on the x-axis. The experimental results are shown with crosses for the mean values and error bars for the minimum and maximum value. For a

negative bending moment (left side of Figure 9), the simulations show a neglectable influence of the coefficient of friction for the ring deformation. For both axial and tangential strain, the results of the FE model fit very well with the experimental results. Only the values of the strain gauge at 30° deviate between simulation and experiment. For a positive bending moment (right side of Figure 9), the coefficient of friction highly influences the strain of the bearing outer ring in the simulation. When comparing the simulation with the experiment, the results vary at different positions. At 120° and 150° for example, the

experiment fits the simulation with a coefficient of friction of 0.2. At 180° the results match when the friction is modelled with a coefficient of 0.15, and at 210° with a coefficient of 0.1. Furthermore, the position of the expected maximum strain deviates. For a pure bending moment, the maximum strain would be expected in the center of the high loaded areas, in case of an My at

0° or 180° as shown in the simulation results. But the maximum strain of the experimental results for a positive bending moment occurs at 150°. However, it is noticeable that these differences are the same for the axial and the tangential strain. That indicates that grease got between the segmented bearing outer rings caused by an opening of the gap that partially reduces the friction. That leads to different states of friction for the flange surfaces of the ring split.

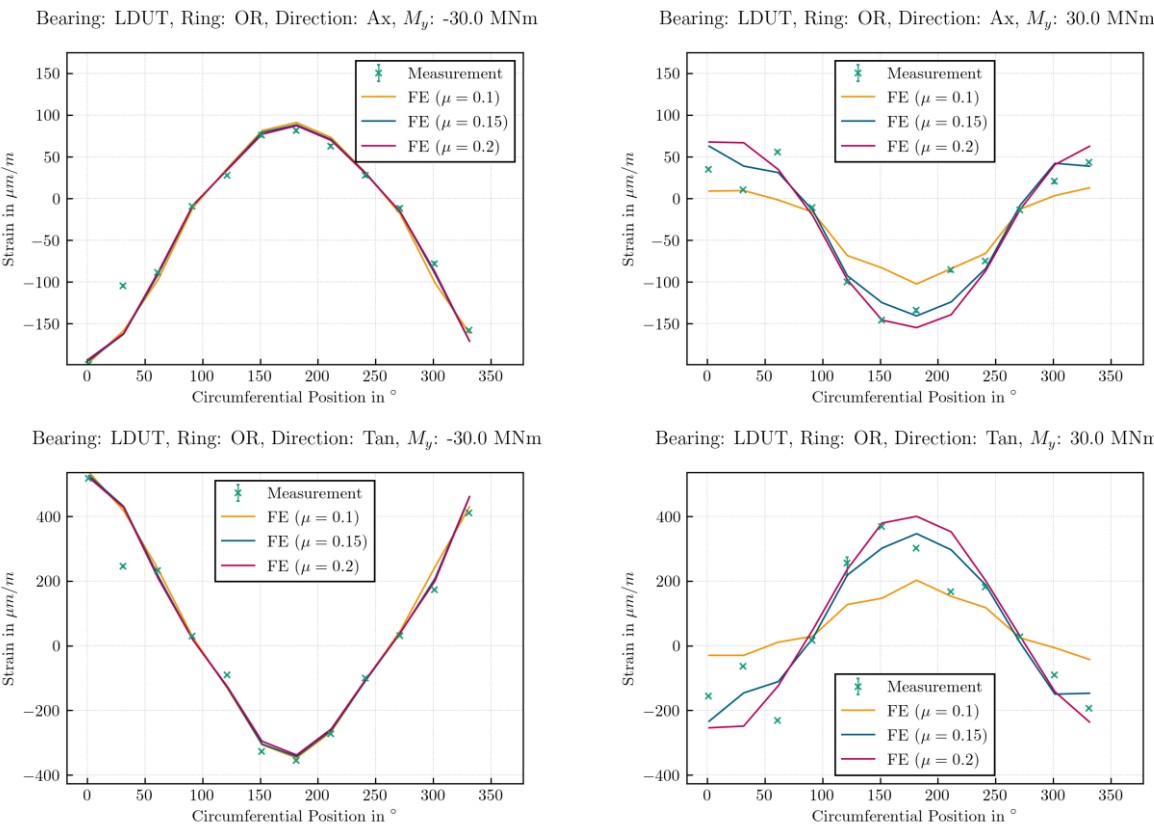

**Figure 9: Axial and tangential strain of the lower bearing outer ring for bolt preload set 1 and My = ±30 MNm with constant bolt preloads of 330 kN**

The general differences of the influence of the coefficient of friction for a negative and positive bending moment can be explained with the geometry of the lower hub adapter shown in Figure 4. When the bearing is loaded with a negative bending moment, the outer ring is pulled away from the stiffer part of the adapter which does not influence the deformation behavior. In contrast, when the bearing is loaded with a positive bending moment, the outer ring is pushed towards the stiffer region of the adapter which leads to a different deformation of the bearing ring that favors radial sliding when the bolt forces are too small to prevent it.

To investigate the influence of different bolt preloads on the strain, the measured standard deviation is subtracted and added to the mean values of the measurement bolts. The variation of bolt preloads is simulated for all three different coefficients of frictions leading to a total of nine simulations. Figure 10 shows the strain on the outer ring considering the standard deviation

of the measured bolt forces in the simulation for a bending moment of My = ±30 MNm with a coefficient of friction of 0.2. The influence of different bolt preloads within the measured standard deviations is significantly less than the influence of the coefficients of friction. The simulation results show nearly no differences in the strain for most parts of the rings. Only around 0° where the stiffer part of the hub adapter is located, the results of the simulations differ slightly.

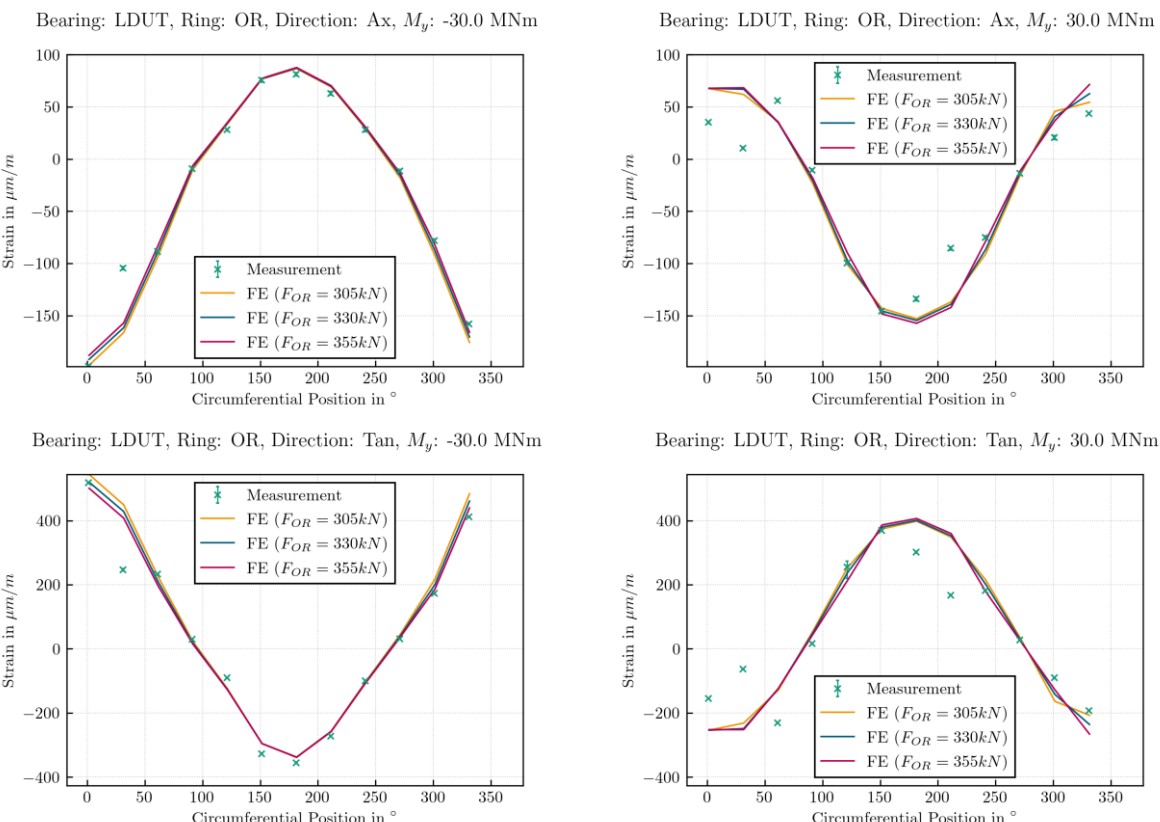

**Figure 10: Axial and tangential strain of the lower bearing outer ring for bolt preload set 1 and My = ±30 MNm with constant coefficient of friction of 0.2**

Figure 11 shows the strain on the inner ring of the lower bearing for a negative bending moment on the left and a positive bending moment on the right. As the segmentation of the rings is only on the outer ring, for the inner ring only the influence of different bolt preloads is investigated. The results of the simulation clearly show no visible changes in the strain of the inner ring when the bolt preloads vary within the measured standard deviation.

The comparison of the simulations with the experiment shows an overall good fit. However, two different phenomena can be observed: Single outliers, where solely one strain gauge value differs and does not fit the characteristic. And deviations in a wider area, where multiple adjacent strain gauge values differ. Single outliers in the experimental data are notable at both inner and outer ring. For the outer ring, the strain gauge at 30° does not fit the expected characteristic of the strain along the bearing circumference and therefore has larger deviations to the simulation results. Figure 9 and Figure 10 both show less axial and tangential strain at 30° for a negative bending moment. In Figure 11 a single outlier is visible at 0° on the inner ring with larger

axial and tangential strain for a negative and positive bending moment. Such discrepancies of single experimental results might have different reasons. On the one hand the strain gauge might be misaligned and glued to the bearing ring with a small angle. As the axial and tangential strain gauges of each position are on the same foil, a misalignment would be noticeable in both directions as shown in the figures. On the other hand, the glue that bonds the strain gauge to the bearing rings might be distributed unevenly under the foil influencing the resulting strain. Differences between experiment and simulation in a wider area can be seen from 150° to 210° for the tangential strain on the inner ring (cf. Figure 11). Those discrepancies might be caused by anomalies on the surface of the surrounding structures. In this case, it would mean there are defects on the flange surface of the FTE that is mounted to the inner ring. The influence of local effects like dents on the bearing behavior have already been studied in (Graßmann et al. 2024). It shows that dents and inclined flanges that are within the manufacturing tolerances greatly influence the deformation behavior of the bearing.

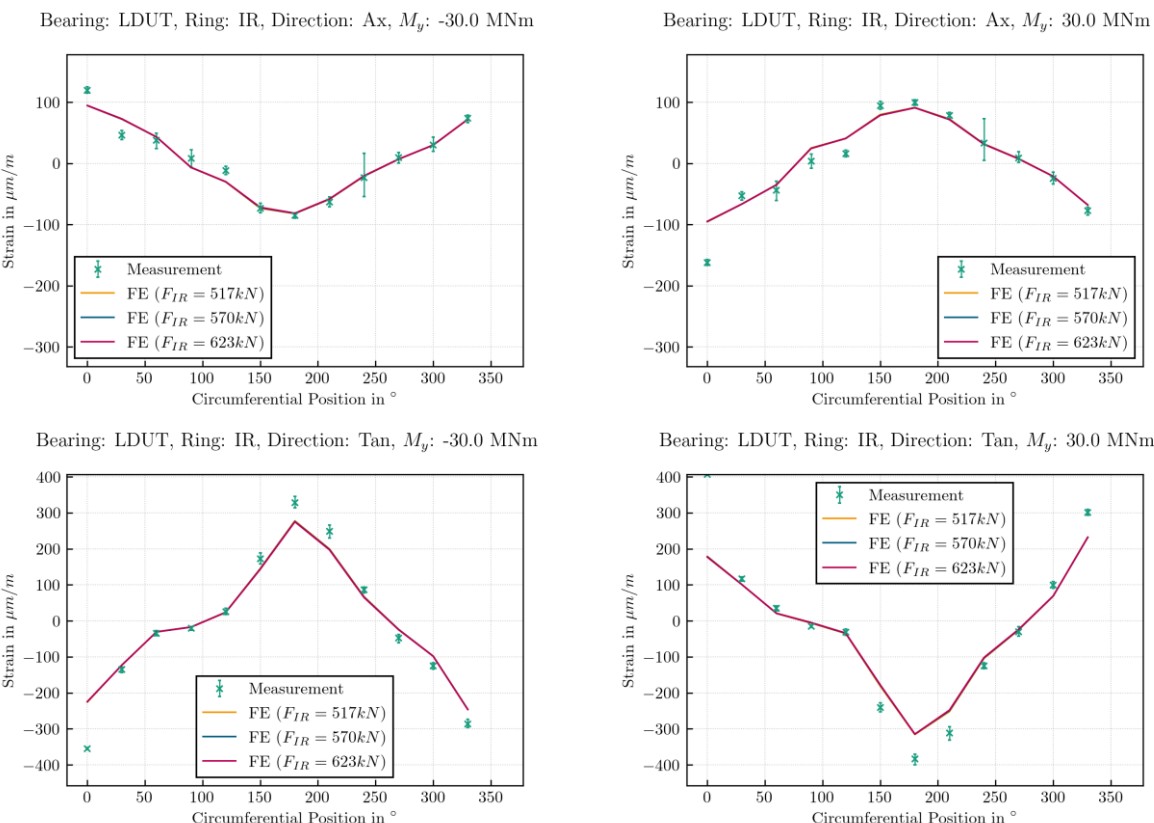

**Figure 11: Axial and tangential strain of the lower bearing inner ring for bolt preload set 1 and My = ±30 MNm with constant coefficient of friction of 0.2**

### 3.2 Bolt preload set 2

For bolt preload set 2, the bolt forces are large enough to prevent any nonlinear effects like separation and sliding in the contact of the bearing outer ring split. As no sliding occurs, different coefficients of friction do not influence the deformation results.

The following results are obtained with a constant coefficient of friction of 0.2 for the outer ring split of the lower bearing. Figure 12 shows the axial and tangential strain on the outer ring for a negative and a positive bending moment of $M_y = \pm 35$ MNm for the different bolt preloads within the measured standard deviation of bolt preload set 2. Again, the comparison of simulation results with the experiment fit very well. The different bolt preloads have only minor effects on the occurring strain and are barely visible in the graphs. The characteristic of the simulated strain matches the experiment with same positions of minima and maxima. Only smaller differences of absolute values occur at the maximum strain for the axial strain of a negative bending moment and tangential strain of a positive bending moment. The single outliers for the strain gauge at 30°, seen on the outer ring with bolt preload set 1, are still visible for a positive bending moment. However, with higher bolt preloads the outliers are far less pronounced.

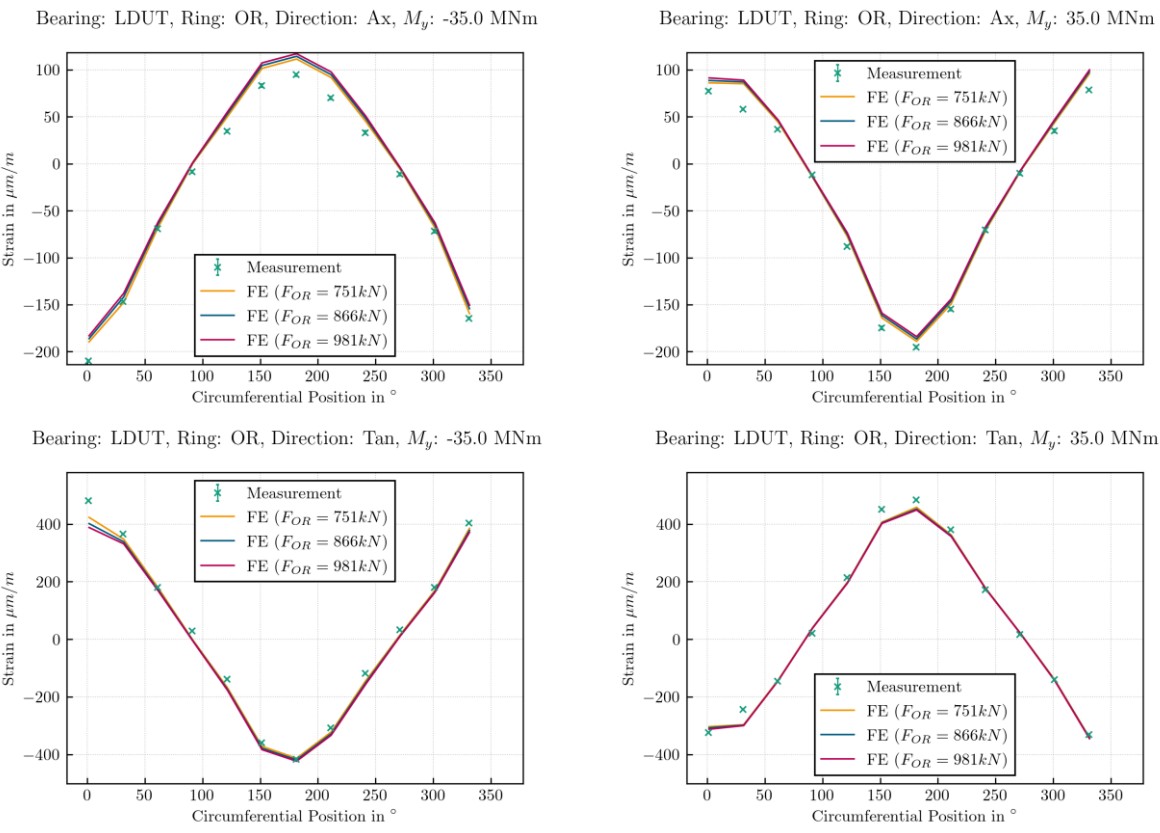

**Figure 12: Axial and tangential strain of the lower bearing outer ring for bolt preload set 2 and My = ±35 MNm with constant coefficient of friction of 0.2**

The results of the strain on the inner ring for different bolt preloads are shown in Figure 13. Again, different bolt preloads within the standard deviation do not lead to visibly different results. As the magnitude of the bolt preloads on the inner ring do not change, the same phenomena than for bolt preload set 1 can be observed. The single outlier at 0° is still very pronounced. Further, the deviations in a wider area between 150° and 210° remain for a positive bending moment especially at the tangential strain. In addition, a shift of the maximum axial strain towards 150° is noticeable for a positive bending moment. For a pure

bending moment, the expected maximum would be at 180°. For a negative bending moment, no larger deviations can be seen.

That strengthens the possibility of an anomaly on the touching surface as the discrepancies only occur for a certain load direction and on a restricted area.

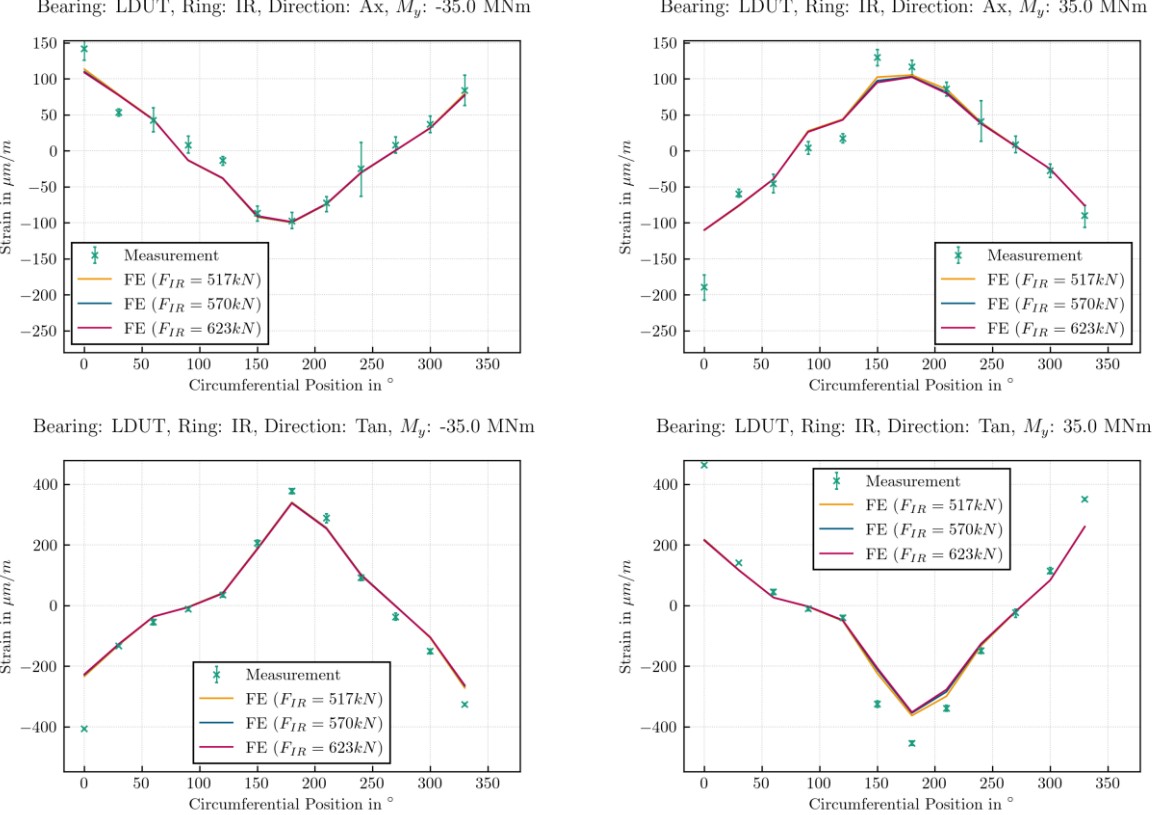

**Figure 13: Axial and tangential strain of the lower bearing inner ring for bolt preload set 2 and My = ±35 MNm with constant coefficient of friction of 0.2**

As stated in the beginning, two criteria must hold true for successful validation: the maximum strain of the FE results should deviate less than 10% and the characteristic course of the FE results should match the experimental mean values. Except for the axial strain for a positive bending moment, the characteristic courses match very well. The simulation and the experimental results match the best with a coefficient of friction of 0.2 and bolt preloads of 751 kN (measured mean value minus standard deviation) for the outer ring and 517 kN (measured mean value minus standard deviation) for the inner ring. Table 3 lists the

maximum strains and the deviation of the simulations from the nearest experimental strain. The minima and maxima are the measured minimum and maximum strain within the 10 seconds of that load level (cf. Section 2.5). The results show less than 10% deviation except for the tangential strain on the inner ring for a positive bending moment. Considering the occurring phenomena explained above, the FE roller bearing and test rig model could be successfully validated for most parts of the bearing.

Table 3: Maximum experimental and simulation strain for bolt preload set 2

| | Direction | My in MNm | Position in ° | Experiment | | Simulation | |
|---|---|---|---|---|---|---|---|
| | | | | Minimum strain in μm/m | Maximum strain in μm/m | Strain in μm/m | Deviation from nearest measurement value in % |
| Inner ring | Axial | -35 | 180 | -109 | -86 | -100 | 0.00 |
| | Axial | +35 | 180 | 106 | 126 | 105 | 0.50 |
| | Tangential | -35 | 180 | 368 | 385 | 340 | 7.58 |
| | Tangential | +35 | 180 | -460 | -446 | -363 | 18.63 |
| Outer ring | Axial | -35 | 0 | -211 | -209 | -190 | 9.27 |
| | Axial | +35 | 180 | -196 | -194 | -189 | 2.58 |
| | Tangential | -35 | 0 | 475 | 477 | 428 | 9.78 |
| | Tangential | +35 | 180 | 483 | 485 | 458 | 5.07 |

Figure 14 shows the radial displacement of the lower bearing outer ring for a coefficient of friction of 0.2 and bolt forces on the inner ring of 570 kN and on the outer ring of 866 kN for the full bending moment ramp from -35 MNm to +35 MNm. It displays the measured displacement and the simulated displacement at 0° on the left and at 180° on the right. All the displacements are nominated to zero load on the bearing to consider any offset in the experimental data and to ensure comparable results. The experimental and simulative results of the radial displacements are very comparable. The largest deviations can be seen at 0° for a positive bending moment. That matches the behaviour of the strain gauges and again shows the influence of the stiffer part of the hub adapter when the bearing rings are pushed towards it.

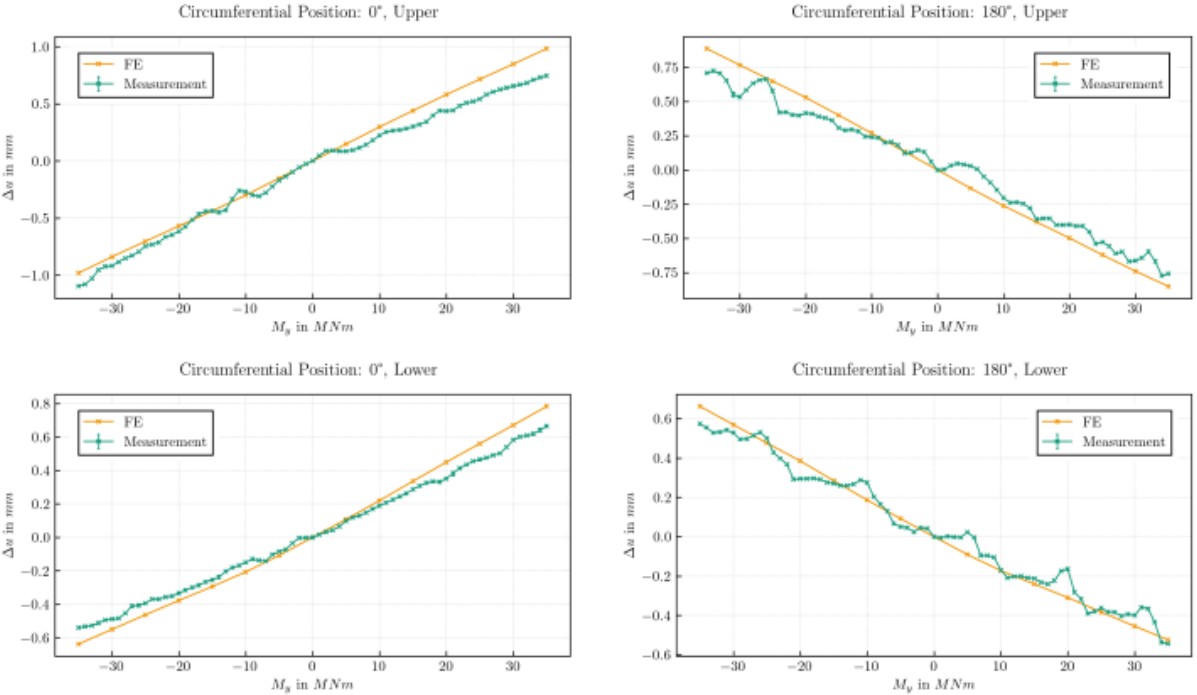

**Figure 14: Radial displacement of the lower bearing outer ring for bolt preload set 2 and My = ±35 MNm**

## 4. Conclusion

To ensure the accuracy of finite-element models, the validation of these models against experimental data is most important. To the knowledge of the authors, large three-row roller bearing models are not publicly validated so far. This work presents a validation method for large roller bearing models. Notably, the behavior of the split ring with possible gap opening and relative motion is a challenge for finite-element analysis. The authors have managed to fulfill the criteria for validation proposed in earlier work. However, single outliers and repeatable deviations over larger areas of the circumference have indicated sensor application and geometrical deviations.

This work has compared the deformation behavior and radial displacements of a finite-element bearing model with experimental data. The aim is to validate the FE model including the bearing and the surrounding structures. The bearing is a 5 m blade bearing of a wind turbine. It is a three-row roller bearing with a segmented outer ring. Nonlinear spring elements represent the roller-raceway interactions. The bearing has been tested on the Fraunhofer IWES large blade bearing test rig BEAT6.1. Specially designed adapters ensure realistic loading and deformation situations of a blade bearing referring to the IWES reference wind turbine IWT-7.5-164. The finite-element test rig model contains every component as in the experiment considering modelled bolts and frictional contacts. For the validation, strain gauges have been used to compare axial and tangential strain at different positions on the bearing rings. Two different bolt preload sets have been considered in the experiment and the simulation. With smaller bolt forces, the outer ring of the bearing has shown gap opening and small sliding between the surfaces of the split ring. Larger bolt forces have prevented any gap opening and sliding. The results have shown significant differences in the behavior of the bearing. Smaller bolt forces have led to the opening of the gap between the two outer ring segments possibly allowing grease to get in the contact area. Therefore, coefficient of frictions might have varied. Varying the coefficient of friction in the finite-element simulations have also indicated different friction conditions along the circumference. Furthermore, the influence of the unsymmetrical hub adapter with a stiffer part around 0° has been shown with the simulations. These influences have not been visible with larger bolt forces showing a more robust behavior of the bearing when sliding and gap opening is restricted. Different bolt preloads within the measured standard deviation have shown only minor effects on the strain of the bearing rings. Single outliers within the experimental strain have been detected. Those outliers can be explained with misaligned strain gauges or inaccuracies with the glue as deviations have been measured for the axial and the tangential strain and for negative and positive bending moments. For the inner ring, the tangential strain has been differed the most between 150° and 210°. With larger bolt preloads, the position of the maximum axial strain on the inner ring for a positive bending moment has been shifted towards 150° leading to a different characteristic of the course. That possibly indicates an anomaly on the surface of attached components as already seen in other works. Taking all these influences on the strain on the bearing rings into account, the characteristics of the courses fit very well and the maximum deviation between simulation and experiment is less than 10%. Therefore, both defined criteria are fulfilled concluding this work with a successful validation of the finite-element bearing model for most parts. In addition, the simulative results of radial displacements of the bearing outer ring at 0° and 180° fit well with the experimental data. That shows that the strain in the bearing rings as well as

the displacements of the bearing rings which correspond to the stiffness of the bearing is accurately represented in the FE model.

This work has shown the influence of different coefficient of frictions for the internal contact between the surfaces of the split ring of a three-row roller bearing. Investigations of different bolt preloads have shown significant influence on gap opening and sliding of the bearing's split outer ring. It further has shown how unpredictable the deformation of the ring becomes when

grease gets in between the segmented ring and highlights the importance of proper tightened bolts for operating roller bearings. Future work will focus on investigating the sensibility of three-row roller bearings on anomalies in the flanges of surrounding structures as well as providing a detailed guideline on the modelling of those bearings.

## Data availability

Data will be made available on request.

## Author contributions

**Matthis Graßmann:** Conceptualization, Methodology, Validation, Writing – Original Draft
**Martin Geibel:** Data curation, Visualization,
**Florian Schleich:** Methodology, Writing - review & editing

## Competing interests

The authors declare that they have no conflict of interest.

## Acknowledgments

The authors gratefully acknowledge the contribution of Dr. Matthias Stammler for internally reviewing this work.

## Financial support

This research has been supported by the German Federal Ministry for Economic Affairs and Climate Action under grant

number 0324344A.

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
