# Peer review of "Validation of a finite-element model of a 5 m three-row roller wind turbine blade bearing"

_Wind Energy Science, 2025_

## Author Comment (AC1)

The following shows the comments of the reviewer. The author's response to each comment is in bold letters.

In this manuscript, the authors present a detailed FE model to simulate the static response of a large three-row roller bearing intended for wind turbine applications. The ultimate goal of the study is the validation of this model, which is achieved by comparing experimental results with those obtained from the FEM, which includes the structure of the test rig itself.

The topic addressed is undoubtedly of great relevance to the wind energy industry, since such models are widely used both in the development and in the certification process of pitch bearings. The scarcity of references on the subject—mainly due to the confidentiality with which manufacturers protect their know-how—fully justifies the interest of this work for researchers and engineers working in the field. The manuscript is also well written and well organized, making it easy to follow and understand.

However, considering that the main objective of the paper is the validation of the FE model, this reviewer thinks that it is not described in sufficient detail for the work to be reproducible and, consequently, for its contribution to be significant enough. In addition, the considerable effort invested by the authors in developing the model, along with the undoubtedly expensive experimental campaign, could be further leveraged to draw more relevant conclusions that would help structural analysts of such components to develop reliable and efficient models.

For these reasons, this reviewer raises the following comments, grouped into two sections.

- \* General comments regarding the scope of the work:
- It would be highly valuable to provide a modelling guideline, offering a more detailed description of the model used and giving recommendations on aspects such as mesh size selection, contact configuration (beyond simply differentiating bonded/frictional types), and other modelling details discussed below.

The aim of this work is to develop a methodology for validating FE models of large roller bearings using a blade bearing test rig. The reviewer is right, to reproduce the bearing model, a more detailed guideline would be helpful. The authors think a detailed guideline along with the validation would lead to an overcrowded and too long paper. The authors plan to provide a guideline in separate work. The defined frictional contacts all use the Augmented Lagrange formulation with default values for contact stiffness and penetration tolerance and the mentioned frictional coefficients in the work. All bonded contacts are using the MPC formulation with default values for contact stiffness and penetration tolerances. The authors have had good experiences with these settings in multiple simulations with different bearing types and surround structures like test rigs and full rotor models. This

information (contact formulation and default settings) can and will also be added to the paper.

As the main outcome of the bearing model is the contact forces derived from the spring elements, different mesh sizes for the bearing rings including the raceway showed no differences in the bearing behavior according to internal investigations.

The virtual strain gauges are the same size as the foil of the experimental strain gauges. Each virtual strain gauge is placed on an element on the bearing rings that has exactly the same size which determines the mesh size of the bearing rings. This information will also be added to the paper.

- Only strain results are compared, although it seems it would have been quite straightforward to also measure displacements, which would allow assessing the model's capability to predict bearing stiffness. While strains are undoubtedly an essential parameter for model validation, the relative deformation between the inner and outer rings is not considered, even though it is strongly influenced by some of the modelling aspects discussed later. A comparison between experimental and simulated displacements would represent a meaningful contribution, without complicating the experimental setup (for instance, this could be easily done using dial gauges). Is there any specific reason why this comparison was not made? Am I missing something?

During the test campaign, radial displacements of the outer bearing ring at the highest loaded areas around 0° and 180° have been measured using laser sensors. The laser sensors are installed at a lever arm that is fixed to the ground which allows for an absolute measurement of the ring displacement. Initially, the authors decided to just focus on the strain gauges as we also did for the validation of other bearing models in the past. However, we are happy to add those results to the paper to add value to the presented work. As the laser sensors only measure a very tiny spot on the rings and the mesh of the bearing is quite course compared to the size of the laser point, multiple nodes that are located near the measured position are used to evaluate a mean displacement in the region of the laser measurement when the FE results are obtained. The following figure as well as a description of the laser measurements will be added to Section 2.4 Measurement.

The following figure shows the results of the radial displacements of the bearing rings at 0° and 180° for loads varying from -35MNm to +35MNm. Both the experimental and the FE results are nominated to 0 load to consider any offset in the experimental data and ensure comparable results similar to the procedure with the strain gauges. The experimental and simulative results of the radial displacements are very comparable. The largest deviations can be seen at 0° for a positive bending moment. That matches the behavior of the strain gauges and again shows the influence of the stiffer part of the hub adapter when the bearing rings are pushed towards it. As the displacements should add value to the model validation, this will only be added for bolt preload set 2. The figure including the discussion of the results will be added to Section 3.2 Bolt preload set 2.

- \* Comments regarding the model description:
- For the simplified modelling of the ball-raceway contact, it is indeed common to use a mechanism similar to the one proposed in this paper, where springs connect the curvature centres of the raceways. However, in the case of rollers, it is more typical to use a spring bed connecting the raceways directly (Golbach 1999) or even a single spring per roller (Kania 2006), without the need for the V-shaped mechanisms shown in Figure 2. Why was this mechanism chosen instead of a spring bed or a single-spring approach? Have different modelling options been tested and compared? It would be very interesting to evaluate different alternatives in terms of accuracy and computational cost, especially in a bearing with such a large number of rollers. Furthermore, why are used 5 or 3 springs per roller, and not more or fewer? Have simulations been carried out in this regard? What conclusions were drawn? Could any guideline be derived on the number of springs to use depending on the roller length or other parameters?

The authors started with simulating ball bearings where the approach of using nonlinear springs that connect to the raceway comes from, as mentioned by the reviewer. With ball bearings, the authors showed that using small parts of the raceway and stiff connections (MPCs) between springs and raceway artificially stiffens the bearing behavior (see <a href="https://doi.org/10.1016/j.finel.2023.103957">https://doi.org/10.1016/j.finel.2023.103957</a>). To overcome that, the authors used larger sections of the raceway and deformable connections (FDCs) and achieved better results. This method was then transferred to roller bearings resulting in the approach of this work. The V-shape indicates that each spring is in contact with a larger section of the raceway to reduce the artificial stiffening. Furthermore, if each spring directly connects the two raceways, they will cause local indents in the mesh which may lead to an unrealistic deformation

behavior of the bearing. The future work of a modeling guideline will compare these different approaches and examine the differences between these.

Regarding the number of springs, WANG et al. (2017) and HE et al. (2018) who are cited in the paper compared the results of a bearing model with solid modeled rollers and one with rollers that are represented by springs. They showed that the minimum of springs to model one roller is 3 to get reasonable results. They got better results by using more springs of up to 8. Because of that, the authors of the present work decided to use the minimum of 3 springs for the radial rollers as there are many of them to save some computational effort, and 5 springs for the axial rollers to have a better distribution of the forces along the roller length as the axial rollers are of main interest. In addition, internal investigations have shown that the bearing behavior regarding roller forces and ring deformation are the same when the rollers are modelled with 5 springs and 31 springs (minimum odd number as stated in ISO 16281 for a discretization of a roller). That investigation is part of a different paper which is currently in the review process and unfortunately cannot be cited right now. However, the information about the internal investigation will be added to the paper.

- Regarding the formulation of the roller–raceway contact, there are several alternatives to the one cited in the paper (Palmgren 1964). More recent formulations, such as those by Puttock (1969), Norden (1973), Tripp (1985), Johnson (1989), Hamrock (1991), or the more recent one by Houpert (2001), could also be considered. Is the Palmgren formulation the most suitable for simulating roller–raceway contact in the case of logarithmic profiles, as studied here? Were other formulations tested?

The approach by Palmgren is used because it is used and referenced to in ISO 16281. For this work, the authors have not compared different formulations for the roller-raceway contact. However, we have compared our models with other research institutes and bearing manufacturers and achieved very comparable results.

Working out the possible influences of different formulations of roller-raceway formulations is a very interesting topic, that will perfectly fit in the modelling guide. Thank you very much for pointing out that topic. Also, the planned investigation will be added to the outlook section of the paper.

- Concerning the implementation of the FDC formulation, and if I understood correctly, each spring is connected to the whole raceway sector that corresponds to each roller. More details on this modelling choice would be required. Why was the spring connected to the whole surface instead of only to the area where the contact is expected (which would be much smaller)? Why not use a rigid-type (MPC/RBE2) spring-raceway connection? Have comparisons been made? Such tests could lead to useful conclusions in terms of accuracy and computational cost.

As mentioned in the response to an earlier comment, using rigid connections will artificially stiffen the raceways which will lead to a too stiff behavior of the bearing rings. The reviewer is right, using a smaller part of the raceway will decrease the number of nodes that are part of the connection and reduce the computational time to some extent. However, when the area that is connected to the springs is too small, we saw local indents in the mesh which led to a less accurate behavior of the bearing. Again, this investigation was carried out for ball bearings and then transferred to the roller bearing model.

- Regarding the other contacts in the model, key details are missing, such as the formulation used in each case, the penetration tolerance or the normal contact stiffness (for penalty-based contacts), and other parameters that may significantly influence the model's behaviour and displacements/stiffness results.

All frictional contacts are using the Augmented Lagrange formulation with default settings for the contact stiffness and penetration tolerance. All bonded contacts use the MPC formulation with default settings as well. This information will be added to the paper.

- As for the mesh, was a mesh sensitivity analysis performed? Were quadratic elements used throughout the model? Could any recommendations be provided regarding mesh size and element type?

Given the large size of the model and the high level of detail including all bolts and frictional contacts, we used linear elements with rather course meshes to achieve a model that can be handled. Only in the contact areas and the bearings itself, the meshed were refined. Please note that despite the strain on the bearing rings, the main goal of this model is not to evaluate stress distributions but the contact forces of the rollers. In addition, internal investigations with our BEAT1.1 (a smaller test rig for 750mm bearings with the same test principle as the BEAT6.1) showed no significant influence on the roller forces and resulting strain on the virtual strain gauges between quadratic and linear elements.

- It is essential to describe the different load steps. Given the nonlinear nature of the model, the order in which loads are applied will influence the results.

On page 9, the authors describe the order of the load steps as the following:

In the simulation of the test rig, the first load step applies gravitational forces and bolts forces. The second load step applies load to the cylinders to compensate for the weight above the lower bearing. This is also done in the experiment and is used as starting point for every test as the lower bearing is unloaded. Following load steps then apply the load combinations that test the bearings.

- Concerning the simulation of the test rig, figures showing the mesh and model details are missing. The description of the test rig model—beyond that of the bearing itself—is rather brief.

Figure 5 in the paper will be changed to only show the experimental test set up. In addition, the following figure will be added to show the overall FE model to give some impressions on the mesh of the components and modelled bolts. Furthermore, the information about the connection of the bolts to modelled nuts or threads using force distributed constraints will be added to the paper.

Most of the above aspects might not have a major impact on the strain results reported in the paper, but they could significantly affect the relative deformations between the rings, i.e. the bearing stiffness, which is a relevant parameter to consider.

Many thanks to the authors for their work. I am sorry for being so meticulous regarding the modelling aspects, but I sincerely believe that the work carried out by the authors is of great value and that the manuscript could be significantly improved by addressing these. I am also confident that the authors will be able to respond to all of them, so I have no doubt that the paper could be published (at least from this reviewer's perspective) once these comments have been taken into consideration.

---

## Author Comment (AC2)

*The following shows the comments of the reviewer. The author's response to each comment is in bold letters.*

In this article, the authors present the experimental validation of a finite element (FE) model for a three-row roller bearing with an outer diameter of 5 meters designed for a wind turbine—an aspect that, as the author rightly points out, has been scarcely documented in the literature to date.

To validate the model, the experimental results are compared with those obtained through FE analysis, which includes modelling the actual test bench (Fraunhofer IWES BEAT 6.1) under representative conditions.

The study also examines the influence of various critical parameters, such as the friction coefficient, bolt preload, and nonlinear effects, on the validation results.

Regarding validation criteria, the author proposes two distinct approaches: first, the maximum deviation must be less than 10%, and second, the trend must match. These criteria are highly relevant, and although each author may define their own validation criteria, in my opinion, a minimum standardized criterion should be established.

The reviewer suggests addressing the following comments:

General suggestions:

- In the context of wind turbines, although both terminologies are accepted, I would recommend using the term pitch bearing in this article instead of blade bearing. The author should consider this as a suggestion; however, I believe it is important to standardize the terminology.

  **The authors agree that a standardized version of using blade bearing or pitch bearing would help a lot. For this work, the authors choose to use the terminus blade bearing as this describes the assembly situation of the bearing while pitch bearing refers to the movement of the bearing in operation. However, one can imagine different bearing designs where the bearing is not only pitching but also performing other movements. If it is fine with editor, we would like to keep the wording as it is.**

- Please improve the resolution of Figure 7.

  **Will do, thank you.**

Specific comments:

- As mentioned earlier, the author defines two criteria for accepting the validation of the model. On what assumptions has the author based the definition of these criteria?

**This is a very good question. The authors firstly introduced those criteria in a paper where they validated a FE ball bearing model (see Validation of a finite-element model of a wind turbine blade bearing - ScienceDirect). There are only very limited works on validating large bearing models publicly available and to the knowledge of the authors none of them define criteria for successful validation. The authors talked to different experts in this field (e.g. bearing manufacturers, professors teaching FEM, and other researchers) and 10% accuracy of the FE model seemed to be commonly acceptable although it was not publicly documented. However, for very small strains on the bearing ring, even a few μm/m differences would result in large percentile errors. Therefore, the authors suggested using only the largest occurring strain to ensure the FE results are within 10% of the experimental results and added the number and location of maxima and minima to ensure the same overall behavior between model and experiment. If the reviewer has additional thoughts on that or even ideas about what other criteria can be introduced for an even better validation, I would be very happy to have a detailed discussion on that topic even beyond this review process. If the reviewer, or any other, is interested in detailed discussions about model validation please contact the corresponding author.**

- In the FE model of the three-row roller bearing, the axial rollers are represented by five springs and the radial rollers by three. Could the author clarify which guideline was followed to determine this number of springs? Additionally, was a sensitivity analysis performed to evaluate the influence of the number of springs? Finally, was this choice supported by references in the literature?

**To the knowledge of the authors, there is no guideline available. Regarding the number of springs, WANG et al. (2017) and HE et al. (2018) who are cited in the paper compared the results of a bearing model with solid modeled rollers and one with rollers that are represented by springs. They showed that the minimum of springs to model one roller is 3 to get reasonable results. They got better results by using more springs of up to 8. Because of that, the authors of the present work decided to use the minimum of 3 springs for the radial rollers as there are many of them to save some computational effort, and 5 springs for the axial rollers to have a better distribution of the forces along the roller length as the axial rollers are of main interest. In addition, internal investigations have shown that the bearing behavior regarding roller forces and ring deformation are the same when the rollers are modelled with 5 springs and 31 springs (minimum odd number as stated in ISO 16281 for a discretization of a roller). That investigation is part of a different paper which is currently in the review process and unfortunately cannot be cited right now. However, the information about the internal investigation will be added to the paper.**

- According to Figure 2, and as I understand it, the contact between the spring and the raceway appears to be defined over a larger area than the actual contact between the roller and the raceway. The entire raceway is divided into green segments, which suggests that there are no areas without contact. Is this interpretation correct? If so, the contact between the spring and the raceway would be greater than in reality. Has the potential impact of this on the results been analysed?

  **Yes, the reviewer is correct, the entire raceway is divided into segments according to the number of rollers and every segment is connected to springs. The authors started simulating ball bearings where the approach of using nonlinear springs that connect to the raceway comes from. With ball bearings, the authors showed that using small parts of the raceway and stiff connections (MPCs) between springs and raceway artificially stiffens the bearing behavior (see https://doi.org/10.1016/j.finel.2023.103957). With small parts of the raceway connected to the springs, the authors saw local indents in the mesh which led to a less accurate behavior of the bearing. To overcome that, the authors used larger sections of the raceway and deformable connections (FDCs) and achieved better results. This method was then transferred to roller bearings resulting in the approach of this work.**

- The non-linear behaviour of the spring elements is controlled by a force–deformation curve derived from analytical calculations. As I understand it, this force–deformation curve is obtained for a cylindrical roller, whereas the actual roller used in the bearing is logarithmic. Could the author clarify how the formulation was modified to account for this difference?

  **The logarithmic profile shapes the surface of the rollers in the range of micrometers to reduce edge loading. These adaptations of the roller geometry are not implemented in the global FE model. The main result of the FE model is spring forces. Internal investigations with 31 springs showed no significant differences for the spring forces whether the profile is considered in the FE model or not. To calculate the pressure distribution along the roller length, the spring forces are then used as input for a half space model.**

- In the finite element model, frictional contacts between the flanges are defined with coefficients of friction of 0.2 and 0.5. Furthermore, the flanges of the bearings toward the surrounding structures are coated to increase the coefficient of friction to 0.67. Could the author explain the assumptions made to define these values? Additionally, was a sensitivity analysis performed regarding these coefficients?

  **The coefficient of friction of 0.2 is used for dry uncoated steel to steel contact. The coefficient of friction of 0.67 results on a specific coating on the**

**bearing surfaces and was tested and provided by the bearing manufacturer. However, this coefficient was tested for coated steel to steel contact. The inner ring of the bearing connects to the GFRP flange of the FTE. To be more conservative with the coefficient of friction, it was slightly reduced to 0.5 for coated steel to GFRP contact. For bolt preload set 1 only the internal contact of the bearing outer ring is exposed to gap opening and radial sliding. For that contact, different coefficients of friction are investigated in the paper to match the experimental results (see Figure 8). All other contacts have no gap opening and no sliding at all. Therefore, the coefficients of friction were not varied. For bolt preload set 2, no sliding in the entire model is occurring.**

- In line 141, the author states that the reaction frame is the bottom white steel structure that connects the rig to the foundation. Could the author clarify how this connection is modelled? What boundary condition has been defined for this connection?

  **For the boundary condition of the reaction frame all degree of freedom for all nodes at the bottom of the reaction frame are locked. In reality, the reaction frame is mounted to the basement through anchors. This is not considered in the FE model. The information about the boundary condition will be added to the paper.**

- In line 149, the author states that in the first step gravitational loads and bolt forces are applied. Has the influence of the pretensioning sequence on the results been analysed?

  **The authors assume, the reviewer is referring to the stepwise tightening of the bearing as in reality not every bolt is tightened at the same time. This has not been investigated yet but might be an interesting topic for further investigations.**

- In line 170, it is stated that the measurement uncertainty is less than 2%, but its potential impact on the results is not analysed. I suggest adding a paragraph discussing the effect on the outcomes.

  **Initially, this work aimed to compare the extent of deviations between the simulation model and the measurement data. Since these deviations partly exceed the ranges attributable to measurement uncertainty, this aspect was not pursued further in the present work; however, it presents a highly interesting topic for future research. The primary factors contributing to the discrepancies between simulation and reality were assumed in the domain of finite element test rig modeling.**

- In general, for the experimental measurements using strain gauges (figures 8-12), the scatter appears to be very low; the difference between maximum and

minimum values across different points is minimal, although in some points the difference is noticeable. Could the author explain the reason for such low variation? Or what typical deviation do we observe across the different points?

**Care was taken to minimize the disturbing effect of electromagnetic interference as much as possible. Therefore, the scatter is on the order of at most a few tens of micrometers per meter. This is very low as a percentage compared to the magnitude values of the measured strains, which can amount to several hundred micrometers per meter. Some measurement points exhibit greater fluctuation. This can have two causes: Either the corresponding strain gauge was more strongly affected by interference sources, or the test rig setup shows greater local deformation behavior at the corresponding location.**

- The results presented in Table 3 correspond to strain gauges. Although it may extend the length of the paper, in my opinion, it would be valuable to also include the results from the other sensors.

  **As also requested from reviewer 1, radial displacements measured by laser sensors will be added to the paper to rely the validation not only on the strain of the bearing rings but also on the deflection. In the opinion of the authors, further measurement results like friction torque or results from inductive sensors would distract the focus of this work.**

I would like to the authors for their work. This is a very interesting and neccesary contribution, as there are currently no references in the literature addressing this topic. By incorporating the suggested changes, I sincerely believe the manuscript will become a much more comprehensive and robust piece of work. Therefore, once the comments have been addressed and clarified, the reviewer considers that the manuscript is suitable for publication.